# Towards Topology-Free Programming for Cyber-Physical Systems with Process-Oriented Paradigm

**DOI:** 10.3390/s23136216

**Published:** 2023-07-07

**Authors:** Vladimir E. Zyubin, Natalia O. Garanina, Igor S. Anureev, Sergey M. Staroletov

**Affiliations:** Institute of Automation and Electrometry SB RAS, Academician Koptyug Ave. 1, 630090 Novosibirsk, Russia

**Keywords:** distributed control systems, control software, process-oriented programming, topology-independent algorithm specifications, hyperprocess

## Abstract

The paper proposes a topology-free specification of distributed control systems by means of a process-oriented programming paradigm. The proposed approach was characterized, on the one hand, by a topologically independent specification of the control algorithm and, on the other hand, by the possibility of using existing formal verification methods by preserving the semantics of a centralized process-oriented program. The paper discusses the advantages of a topologically independent specification of distributed control systems, outlines the features of control software, argues why the use of a process-oriented approach to the development of the automation of cyber-physical systems is suitable for solving these problems, describes a general scheme for implementing a distributed control system according to a process-oriented specification, and proposes a formal heuristic algorithm for partitioning a sequential process-oriented program into independent clusters. We illustrate our algorithm with bottle-filling and sluice case studies.

## 1. Introduction and Motivation

Cyber-physical systems (CPSs) are usually defined as network systems with a computational core that interacts with the physical world through sensors and actuators. Some researchers insist on adding the network architecture to the definition. This remark is confirmed by the widespread use of distributed architectures for the implementation of cyber-physical systems. For example, a modern car control system can include up to a hundred microcontrollers connected to the CAN bus [1]. The distributed architecture of the vehicle control system provides a reduction in the length and weight of connecting wires, ease of maintenance, eliminates the mechanical or hydraulic implementation of various functions, thereby minimizing the weight and size characteristics of the system as a whole, and also makes it possible to implement various control strategies [2,3,4]. Due to these circumstances, software development has a significant part in the costs of creating new and redesigning existing control systems. Usually, software development costs account for more than half of all expenses [5].

Cyber-physical systems (CPSs) are being actively developed as part of the Industry 4.0 concept. According to the concept, which declares the shift from mass production to mass customization, the CPSs must meet requirements such as modularity, adaptability, reusability, and flexibility. The class of cyber-physical systems that integrate physical and virtual components in the automation of production processes is called cyber-physical production systems (CPPSs). IEC 61131-3 languages and a later superstructure over them—IEC 61499 function blocks—are positioned as the main means of specifying the algorithms for the functioning of CPPSs. These approaches are seriously criticized by specialists for their inability to respond to the challenges that arise in the development of industrial automation systems and, in fact, for their inability to solve the problems for which these languages were created: to ensure portability, reconfigurability, interoperability, and the distribution of the developed software [6]. The developers of the IEC 61131-3/IEC 61499 specifications themselves agree with the conclusions about the standard’s inconsistency with the declared goals [7].

The problems of existing approaches lead to the fact that, in practice, a module-oriented (device-centric) approach is used [8], which involves the separate programming of each controller of the control system. In particular, this is caused by the current “one function per module” paradigm used in the development of control system components in accordance with the V-cycle technology [9]. The main disadvantages of this approach are the complexity of software maintenance, its readability and modification, flexibility, and the complexity of verification, which is crucial for most cyber-physical systems. Based on the current problems of programming cyber-physical systems, the need for the development of the topologically independent (application-centric) programming of distributed control systems was stated in [9]. The indisputable advantage of this concept is the possibility of a simple and, in the most preferable case, sequential description of the control algorithm, independent from the type of nodes, their number, and the topology of the distributed control system. Topology-free programming could provide a drastic reduction in the cost of developing and maintaining the control software being created. The implementation of the concept involves creating a toolchain for the design and development of the control algorithm: first of all, automatically dividing the algorithm into parts and deploying them on controllers. The concept weeds out the tedious and error-prone manual process of dividing it into parallel (more precisely, independent or loosely connected) parts. Although the quality of automatic parallelization has improved over the past few decades, the fully automatic parallelization of sequential programs by means of compilers remains a significant challenge due to the need for complex program analysis and unknown factors (such as input range) at compilation time [10]. This gives rise to serious problems associated with the automatic generation of executable code. Automatic parallelization by compilers or tools is very difficult due to different types of race conditions. Dependency analysis is not easy for code that uses indirection, pointers, recursion, and indirect function calls. In addition, access to shared or global resources can lead to bottlenecks, resource starvation, or deadlocks [11]. The main direction for increasing the comprehensibility and reducing the complexity of the verification of a system of concurrent processes is to ensure the determinism of its execution [12].

These circumstances are typical for the field of parallel programming of computational algorithms. In terms of control algorithms, the complexity of the problem can be reduced due to the following features:First of all, in the field of automation, parallelism is an integral part of the control algorithm. This circumstance is reflected in so-called process-oriented languages; for example, in the recently developed poST language, which is a process-oriented dialect of the IEC 61131-3 Structured Text [13]. In the poST language, a program is built as a set of weakly dependent processes—FSM-like structures. Therefore, the process-oriented specification of the control algorithm contains information about its possible parallelization.Secondly, in contrast to parallel programming goals, the need for parallelization is primarily caused not by the desire to reduce the computation time but by the desire to reduce the cost of the system as a whole and the cost of its maintenance by using smaller, less expensive, and more flexible microprocessors, reducing the length of wires, reducing the complexity of wiring, and improving maintainability, providing the opportunity to implement various control strategies [3,14].Thirdly, for control problems, an extremely important circumstance is the possibility of formally verifying the algorithms being created. For the process-oriented paradigm, we developed the mathematical model of hyperprocesses. Informally, a hyperprocess involves expanding the processes in the system into one process by periodically turning on the logical processes in the program according to the round robin strategy to execute the code in their current states. A number of formal methods for the process-oriented programming languages have been developed [15,16] based on the semantics of a hyperprocess. Correspondingly, when developing a methodology for the topologically independent programming of control systems based on a process-oriented approach, it is desirable to reuse the existing verification methods.

Following the above reasoning, we propose a fairly simple technique for parallelizing the control algorithm on the basis of the process-oriented approach. Our automatic partitioning algorithm works at the process level. We also need to preserve the semantics of the hyperprocess.

The outline of this paper is as follows. Section 2 presents related works on distributed control systems’ design and implementation. The main contribution is presented in the three following parts. First, we describe the main basics of process-oriented programming and formulate the hypothesis of the research (Section 3). The second part contains the general approach for partitioning a centralized process-oriented control algorithm specification into a distributed architecture (Section 4). Finally, we empirically demonstrate the approach using bottle filling and sluice distributed control systems (Section 5). In the concluding Section 6, we discuss the paper’s contribution and outline possible development paths.

## 2. Related Works

### 2.1. Distributed Control Systems for Vehicles

Probably the most well-known area of the application of distributed control today is the automotive domain. Here, the associated data transmission lines are used to connect the main electronic control unit (ECU) of the car to child ECUs and sensors/actuators. Based on this approach, other systems in different domains are being developed for cases where there is a sufficiently large number of interacting devices and it is required to minimize the number of wires. Since the 90s of the last century, the advent of injection technology and the implementation of closed-loop engine control algorithms have led to the need to take into account the data and synchronization of a large number of signals from sensors connected at different places in the system. Thus, the need for distributed control was recognized early enough in the automotive industry and a number of connectivity protocols were developed to enable in-vehicle network communication. Nowadays, these protocols have been merged into a de facto standard for the in-vehicle controller area network (CAN) [17]. A CAN bus has been successfully integrated into the industry for data exchange between controllers in a distributed system with a distance of several meters.

In [18], the authors explored CAN messages from a vehicular peripheral bus of a real car. It was found that the peripheral bus is disconnected from the main bus, where data are transmitted in the exchange of devices for monitoring engine parameters, is less high-speed, and transmits data between the radio, heating installation, display, and control knobs. There are several identifiers and data bytes are transmitted for each identifier with some given periods. From such experiments, the distributed control approach used in industry today can be stated as: (1) controllers transmit periodic messages to each other about the state of key system variables (encoded information received from closely connected sensors and derived information based on the processing of this data); (2) more important information is transmitted at shorter intervals; (3) periods are calculated so that there is not a large number of collisions during simultaneous transmission; (4) the data network is essentially distributed into independent subnets that do not interfere with each other when transmitting heterogeneous information (as in the example, the engine ECU does not need to know about the radio station currently playing). In the existing literature, such a control algorithm is called “time-triggered control” [19].

In [20], Albert et al. compared the applications of the time-triggered and event-triggered approaches in automotive distributed control systems using the CAN bus and its TTCAN variant (time-triggered CAN) [21]. It is noted that systems whose subsystems’ operation is pre-calculated by time windows are predictable; their components can be interchanged from different vendors, with the requirements that the maximum operating times are not violated. However, such systems cannot properly respond to important (system critical state) messages because the latter can only be processed after bus arbitration. On the other hand, fully event-driven systems can process messages with the desired priorities; however, due to non-determinism, when changing system components to others (for example, faster or slower), the control algorithm must be completely revised. It is concluded that it is necessary to use the event-triggered approach only for a few of the most important messages and to include their processing in a special time window before everyone else, while the processing of the normal messages should be performed using time-triggered control. In this case, it is possible to calculate the change in the control law of the object, and it becomes predictable.

Thus, when planning control algorithms, the delays associated with the bus should be calculated and simulated. For instance, the work by Baek et al. [22] considers the theory and practice of creating a controller for an autonomous all-terrain vehicle using the CAN bus. Using a simple fixed-priority bus model [23], the worst-case response time was calculated and the transit times and loss percentages between key system components were measured. Taking into account the obtained information, a simulation was carried out on the controller model based on control theory and it was concluded that the implementation is built correctly, taking into account the specified delays and losses. The same has been carried out in numerous research projects; for example [24,25].

Of course, if the controllers are placed side by side, it makes no sense to use the CAN bus. For example, Shiau et al. explored the possibility of designing a distributed multi-MCU-based flight control system for unmanned aerial vehicles [26]. The system was built on low-cost microcontrollers. For inter-node communication, the UART interface was used. The distributed system is able to calculate complex non-linear Kalman filters and calculate the attitude angle using base elements with a low performance. The system also provides for error handling when receiving sensor data. The quality of the resulting design was verified using MATLAB.

### 2.2. Design of Distributed Control Systems Based on IEC 61131-3/61499

Christensen in [27] suggested and adopted the model–view–control (MVC) design pattern to the domain of industrial automation and integrated it with the IEC 61499 standard architecture. According to the adopted pattern, control software is organized from two components connected in a closed loop: a controller, implementing a set of control operations available as published services, and a model, simulating the plant. Vyatkin et al. in [28] extended the approach to include the formal verification of function block systems, appropriate for more rigorous verification by means of model checking.

Thramboulidis in [29] adopted a system-based approach for the development of industrial automation systems (IASs). Based on the proposed methodology, the UML software model of the system was obtained by extracting it from a formal system model and was further developed into implementation code. The approach presented by the author enables the development of control systems from diverse perspectives to meet the demands of CPSs. The author provided evidence of the need for such an approach by demonstrating the simultaneous engineering of various components of the system, including mechanical, electrical, and software parts. This approach offers the advantage of integrating the relationships between the physical and software aspects of components in CPS, thereby improving the quality of IAS development.

In [30], Schwab et al. described the TORERO tool, which achieved a semi-automatic allocation of the IEC 1499 control application to the distributed hardware underneath, and generated communication-related code automatically based on the allocation.

In [31], Dai and Vyatkin proposed three different approaches to redesigning existing PLC programs to the distributed architecture based on the IEC 61499 functional blocks: two object-oriented approaches and a service-oriented one (class-oriented approach), in which the program is structured based on the plant functionality. A study of the effectiveness of these approaches on an airport baggage handling system showed the improved effectiveness of the latter approach, even when compared to the original IEC 61131-3 program.

Ribeiro and Bjorkman in [32] analyzed and identified several fundamental challenges that need to be addressed before one can start to design cyber-physical production systems consistently. Among other things, the authors stated the existence of two different approaches to the decomposition (structuring) of the system: functional or structural (object-oriented) decomposition. In addition, the authors, analyzing the concept of holonic manufacturing systems, noted the lack of dedicated domain-specific language, leading to agent or service-based architectures or a combination of both in articulation with other technologies.

Patil et al. in [33] proposed refined design patterns for cyber-physical system architecture. These patterns were empirically tested in a series of projects and showed a reduction in development complexity [34].

In [35], Cruz Salazar et al. proposed a classification of multi-agent approaches to the design and implementation of cyber-physical production systems, classified them, and showed their satisfactory properties, particularly for the implementation of the field control level. The analytical part of the work provides an excellent overview of multi-agent technologies that are alternative to the conventional one based on the IEC 61131-3/61499 standard.

Zyubin and Rozov in [36] proposed a conceptual approach to the IEC 61499-based specification for control software using the poST language. The approach is based on the idea of a single reduced function block with one event input invoking the algorithm specified in poST. The approach allows for developing distributed event-driven algorithms using only one ST-like language, and thus drastically simplifies program maintenance. The question of ensuring determinism was left out of consideration.

In [37], the researchers concluded that managing scenarios for reconfiguration within the function block can be difficult at this stage as the reconfiguration and control models are merged into a single ECC. This results in overlapped error handling, initialization, and reconfiguration. Additionally, there is no differentiation between the control level and higher levels within the ECC. This leads to a large number of states and transitions and low readability and maintainability. Furthermore, due to the high dependency between the states, it is challenging for the developer to extend or maintain an ECC.

Sinha et al. in [38] proposed a hierarchical and concurrent ECC (HCECC) to model concurrent behaviors within the IEC 61499 paradigm. The HCECC-based function blocks utilize a multilevel hierarchical state machine to integrate concurrent and hierarchical behaviors and reduce system complexity. However, the authors did not tackle the reconfiguration issue, which is critical in distributed control systems where execution semantics are influenced by environmental changes and user requirements.

Marschall et al. in [39] proposed an agent-controlled approach for CPPSs. Open Platform Communications Unified Architecture (OPC UA) was selected as the standard protocol for the data exchange in the multiagent control system. The internal control logics of the system were implemented using the state machine design pattern. When implementing the system, no verification tools were used and the resulting program contained errors during execution.

Despite the fact that the IEC 61499 standard aims to achieve interoperability and portability, the results of the experiments show that the options for exchanging data between different tools within this standard are restricted. As part of their efforts to tackle the challenge of porting IEC 1499 applications, Hopsu et al. in [40] suggested streamlining manual processes for relocating functional blocks in a distributed system by creating a standalone application in order to use the NxtStudio package, which automatically manages communication between different devices.

In paper [41], the authors proposed an IEC-61499-based model for CPSs to distribute the complexity of control software over numerous small devices. The approach enables the creation of a comprehensive structure that can combine the design, simulation, and distributed deployment of automation software. The proposed scheme was validated through a packet-sorting system implemented in the nxtSTUDIO platform. Thus, the article outlines one of the variants of the so-called device-centric approach.

Parant et al. in [42] proposed a methodology for building the knowledge base from the product’s specifications and a formal diagram linkage table (DLT) approach to ensure the domains’ coherence and interdependence. This approach facilitates the identification of a perturbation’s impact on the system and proposes the appropriate response during a reconfiguration of its functionality. In fact, the authors extended the functional approach to designing CPPSs and included in the development process not only the question “what the system should do” but also the question “what the system can do”.

### 2.3. Solving Conflicts in Distributed Systems

Well-known approaches that deal with non-deterministic behavior during conflicts over shared resources have been developed for concurrent processes. The development of these methods is traditionally associated with Dijkstra’s classic work, e.g., [43]. Reviews of the state of the art in this area can be found in [44]. However, since the concept makes stronger demands and has specifics, particularly the interaction with the environment, these algorithms are only partially applicable to the tasks of creating distributed automation systems.

Eidson et al. introduced a programming model that captures the physical notion of time for the model-based design of distributed real-time embedded systems, called programming temporally integrated distributed embedded systems (PTIDESs). PTIDES structures distributed software as an interconnection of components communicating using timestamped events in order to provide determinism in CPSs [45].

Among the works on modeling cyber-physical systems, the papers of Edward Lee should be especially noted. In his book [46] and also in his article [12], he dwells in detail on aspects of the correct definition of the concept of a cyber-physical system. He notes that this concept was introduced by Helen Gill in 2006 and is just a modern rethinking of cybernetic systems, which, in turn, are systems studied in control theory. However, modern cyber-physical systems are associated with large amounts of information from heterogeneous sensors. Such systems operate in a distributed manner and have an impact on people, and, when designing them, modeling cannot be dispensed with, since their correct functioning can have consequences for the environment and humans. In their works, Lee raises the key problem of modeling a non-deterministic world based on different types of models, both deterministic and non-deterministic. For instance, in [12], he gives a detailed description of PRET and Ptides projects, which aimed to use deterministic models for CPS with faithful physical realizations. The author argues that such an approach is practical due to deterministic models being easier to understand and analyze. The PRET project shows that the timing precision of synchronous digital logic can be practically made available at the software level of abstraction. The Ptides project shows that deterministic models for distributed cyber-physical systems have practical faithful realizations. The time stamp mechanism proposed in these projects allows for detecting synchronism violations due to network delay or the clock synchronization error and processing these violations in accordance with the context; for example, rejecting a database transaction.

In [47], in order to avoid undesirable non-determinism in the system of parallel processes (e.g., data racing), M. Lohstroh et al. suggested time-tagging actions of parallel processes that are sensitive to non-deterministic execution. These tags are used in additional timing constraints that are supposed to ensure that concurrent processes run in a consistent manner that does not allow for undesirable non-determinism. This technique requires a careful study of the sequence of the processes’ execution, which is quite labor-intensive.

In paper [48], H. Li et al. studied the possibilities of ensuring the robustness for a distributed consumption–production system with respect to deadlocks. They proposed a strict proxy communication protocol for the system agents (processes) that limits the number and frequency of process interactions. On the one hand, such restrictions strongly lead to the absence of deadlocks in the system, but, on the other hand, they significantly complicate the interaction of parallel processes.

In the present work, however, we propose an algorithm for distributing components into clusters running on the same controller so that synchronization does not have to be taken care of inside of the cluster.

## 3. Theoretical Framework for the Proposed Approach

In this section, we present the basic theoretical grounds for process-oriented programming and describe general schema for the deployment of process-oriented specification on an arbitrary distributed architecture.

When we think about a contemporary control system, we typically envision a digital controller linked to a controlled environment, which includes hardware and equipment where physical processes occur. This environment, known as the plant, is the external element of the control system. Sensors and actuators facilitate the connection between the plant and controller. The sensors collect data from the environment and transmit them to the control system. Subsequently, the controller reacts to the data inputs by generating control values for the actuators. These actuators then modify the flow of physical processes within the plant.

We consider the following features of an industrial automation system fundamental to the class of control programs:interaction with an external environment via sensors, actuators, controls, and indicators;indefinite running time;cyclic execution;event-driven behavior;synchronism, expressed in the active use of operations with time intervals, which is required to ensure that the control program matches the dynamic characteristics of the plant;control flow concurrency, which aims to describe the parallelism of physical processes on the plant;hierarchical structure.

Another important point for the development of a practically useful methodology in the field of creating control programs is the urgent need to ensure structural conformity between the architecture of control programs and the technological description of the plant. Modern studies show that this problem is actually very acute to date [49]. Control software development fits well into the “client–contractor” paradigm. At the initial stages of the project (system requirements specification, program specification), the client (plant and process engineers) plays a leading and irremovable role. Their input gradually decreases as the project progresses to the implementation stage. The contractor (programmer), however, plays an auxiliary and dependent role at the start. It is only at the design and implementation stages of the project that they start to gain relative independence. However, this connection between the client and the contractor remains throughout the entire life cycle, which involves program changes.

The authors strongly believe that a careful consideration of the plant design process allows us to answer the frequently raised question of choosing between a functional decomposition vs. an object decomposition of a control program [50,51,52], to name a few.

The paper [53] argues in favor of the functional structuring of control algorithms. The author draws attention to the fact that the control algorithm appears at the stage of creating a plant. The control algorithm is aimed at the implementation of certain technology and has a process nature. From the point of view of the technological description, the functions performed by the plant and the processes that take place on the plant are more important, while the issue of their object implementation is in the background. For example, the process of heating a certain volume required by a technology can be implemented in many ways electrically, by current and resistance, by heated steam in a pipeline, by a microwave method, by a laser, or by mechanical friction. During the development process, several options are usually considered by plant engineers; moreover, the method of heating the volume (object representation) can change during the operation of the plant. Thus, the process structure of the plant is more resistant to changes and it is preferable to choose it as the base one.

### 3.1. Process-Oriented Programming

Process-oriented programming intends to provide a conceptual consistency of the PLC source code with a technological description of the plant operating procedure. The concept can be seamlessly implemented as a textual programming language for complex PLC software. The approach uses the advantages. The process-oriented paradigm assumes that a control program specified as a set of weakly connected concurrent processes structurally and functionally corresponds to the technological description of the plant. Each process is specified by a sequential set of states. The states are specified by a set of arithmetic constructs, extended by a TIMEOUT operation, SET STATE operation, and START/STOP/check state operations to communicate with other processes.

The process-oriented paradigm uses the hyperprocess model to represent the control program that is an ordered set of processes, which are cyclically activated with a period in a predetermined order. A formal definition of the model can be found in [13]. At this point, we can assume that a process is just a function or a set of instructions (in a programming sense). It should also be noted that a kind of perfect synchrony hypothesis [54] is assumed in this model.

The newly developed poST language is one of the process-oriented programming languages that implement the concept. The poST language can be utilized seamlessly as a textual programming language for intricate PLC software in the IEC 61131-3 (3rd Edition) context. The language incorporates the benefits of FSM-based programming and the traditional syntax of the ST language, making it easy to adopt. According to the poST language, a poST program is a collection of weakly connected concurrent processes that structurally and functionally correspond to the plant’s technological description. Each process is defined by a sequential set of states, and each state is specified by a set of ST constructs that are extended by a TIMEOUT operation, SET STATE operation, and START/STOP/check state operations to communicate with other processes. The paper [13] outlines the fundamental syntax of the poST language and demonstrates the usage of the poST language in creating control software.

### 3.2. Deploying a Process-Oriented Specification on a Distributed Architecture with Semantics of a Centrally Controlled Implementation

The main idea of mapping a process-oriented algorithm to a distributed architecture is shown in Figure 1. The mapping preserves the semantics of a hyperprocess since the algorithm is in fact executed sequentially. In accordance with this scheme, the execution of a process-oriented algorithm occurs cyclically while maintaining the “read–calculate–write” cycle. The reading of the input signals is performed in parallel, and then the cycle of the inter-node synchronization of the read values is performed. After the synchronization cycle, the processes on the first computing node are activated. At the end of the calculations, the control (interprocess) signals and the calculated values of the output signals are transmitted to the stakeholder nodes; at the end of the synchronization, the second (in the general case, to the next) node computer executes its part of the control algorithm. After the sequential execution of the algorithm on all computing nodes of the distributed control system, the last microprocessor node initiates a parallel writing of output signals. After this, the next cycle begins.

The discussed approach enables us to generate a distributed application for any topology, comprising nodes up to the number of processes in the original process-oriented specification. When specifying the control algorithm, we do not care about the number of nodes, their type, and the network topology. This makes our approach topology-free programming.

Advantages of the approach:deterministic behavior of a distributed system;a high degree of granularity up to placing only one process on a node of the distributed system;preserving the semantics of the monolithic implementation.

Besides the advantages, the designed model also leads to a more intricate software architecture as only one PLC can have access to a single output module. It also requires additional communication between multiple PLCs to share information such as input/output module data and program data. As a result, there is an increase in both hardware costs and engineering effort required [31].

## 4. Proposed Approach

### 4.1. Motivation and Hypothesis

Summing up the above, we can state that a deterministic model is a beneficial property for a system as it defines exactly one behavior under the given inputs. This property is useful as it can establish the accurate behavior of the system being modeled, given the same inputs. This type of model is helpful for developing tests to determine if a physical system conforms to the model and is ready for deployment. However, a nondeterministic model is less useful for this purpose as it can have multiple possible “correct” behaviors [12]. Thus, deterministic systems are easier to understand and analyze. They are easier to validate and hence easier to certify.

However, distributed architectures suffer from several significant limitations, such as the possibility of conflict arising from multiple nodes having write access to outputs. To address the problem, IEC 61131-3 specifies that every remote output module should either be allocated entirely to a single PLC, or a critical section/token mechanism should be employed that restricts access to a single PLC at any given time. While this was originally intended to prevent non-deterministic behavior and guarantee data synchronicity, the downside is that, at the design phase, the owner of each output module must be predetermined, leading to a decrease in runtime flexibility [31] and forcing the developer to use the device-centric paradigm. The lack of structure in granularity can result in interactions between components that operate at different levels of abstraction. These parallel communication channels, which can operate asynchronously, have the potential to introduce inconsistencies in the overall behavior of the system [32].

The theoretical section indicates that a process-oriented specification, specifically in the poST language, includes a loosely coupled process algorithm that resembles critical sections in parallel programming. Given this information, we propose a hypothesis that suggests using the poST program as an architecture-independent specification for a distributed control system. The individual processes of this specification can be combined into groups or “clusters” following specific rules. The clusters, in turn, can be executed on the nodes of a distributed system, and the deterministic behavior of the system as a whole can be provided with minimal or no synchronization effort.

In summary of the subsection, the objective of dividing a process-oriented specification into clusters is to identify sets of processes that are closely interconnected internally while being only loosely linked to the rest of the system.

### 4.2. The Heuristic Approach

To minimize the computational complexity of inter-node synchronization, which should ideally be excluded, we propose to divide the set of program processes into independent groups (clusters) with the finest possible granularity. The heuristic requirements for partitioning a control algorithm according to a process-oriented specification into clusters are as follows:two processes using the same variable should both be in the same cluster;two processes using the same process should both be in the same cluster;processes forming a loop relative to the use relation should be in the same cluster.

Let the letters *p*, *v*, and *e* (possibly indexed) denote processes, variables, and elements (both processes and variables), respectively. In poST program *u*, expression pusee denotes that *p* uses *e* in program *u*, i.e., the specification of *p* includes operations with *e*. A transitive closure of the relation use bounded by processes only is $use^{+}$.

We define the relation ∼ on processes of program *u* as follows:If $p_{1}\;use\;e$, and $p_{2}\;use\;e$, then $p_{1}∼p_{2}$.If $p_{1}\;use^{+}\;p_{2}$, and $p_{2}\;use^{+}\;p_{1}$, then $p_{1}∼p_{2}$.p∼p.If $p_{1}∼p_{2}$, and $p_{2}∼p_{3}$, then $p_{1}∼p_{3}$.

The relation ∼ is an equivalence; hence, it divides processes of program *u* into equivalence classes called *clusters*. We can consider a poST program as a sequence of processes. This sequence sets the order on the set of program processes. In reverse, some order on a given set of processes generates a program. Processes in every cluster are totally ordered with respect to the order of the original program. We would like to obtain such ∼-partitioning of the set of program processes that we can order its clusters in a way that the resulting order of processes would be the same as in the original program. The following straight algorithm may produce such partitioning.

Let *n* be a number of processes of program *u*. UE(i) is a set of elements used by process $p_{i}$, and $UP^{*}\left(i\right)$ is a set $\left\{j\;|\;1\;\leq j\leq n\wedge p_{i}\;use^{+}\;p_{j}\right\}$, which is a set of processes used by process *i* possibly through other processes. Let $CLS_{id}\subseteq \left\{1,\ldots ,n\right\}$ be a set of cluster identifiers. Let cl be an array of length *n* such that cl[i]=j means that process $p_{i}$ belongs to cluster j∈CLS.

The array cl implements mapping processes to clusters. The following partitioning algorithm produces the processes clusters for program *u* by checking if every pair of program processes is related by ∼:$$\begin{array}{c} 1.\;CLS_{id}\;:=\;\left\{1,\ldots ,n\right\};\\ 2.\;\mathrm{for}\;i\;=\;1\;\mathrm{to}\;n\;\mathrm{do}\;cl[i]\;:=\;i;\\ 3.\;\mathrm{for}\;i\;=\;1\;\mathrm{to}\;n\;\mathrm{do}\\ 4.\;\;\mathrm{for}\;j\;=\;i+1\;\mathrm{to}\;n\;\mathrm{do}\\ 5.\;\;\;\mathrm{if}\;(cl\left[i\right]\neq cl\left[j\right]\wedge \left(UE\left(i\right)\cap UE\left(j\right)\neq \emptyset \vee i\in UP^{*}\left(j\right)\wedge j\in UP^{*}\left(i\right)\right))\\ 6.\;\;\;\mathrm{then}\;\{cl\left[j\right]\;:=\;cl\left[i\right];\;CLS_{id}\;:=\;CLS_{id}\setminus \left\{j\right\};\} \end{array}$$

Let us explain what happens in the algorithm. The first two lines are the initialization: in line 1, we define the initial set of cluster identifiers to be {1,…,n}, which is the maximal set of clusters for *n* processes, and, in line 2, every process is associated with its initial cluster. Lines 3 and 4 organize pairing cycles for processes to check the possibility of their joining into one cluster. Line 5 checks the joining condition: two processes must not be in the same cluster yet, and they must directly use at least one the same variable (or process) or indirectly use the same processes, i.e., use them through the using process chain. If this join condition is met, then in line 6 we associate the higher numbered process with the cluster ID of the lower numbered process and remove the higher numbered cluster from further processing. Figure 2 demonstrates the algorithm flowchart.

We consider the resulting partitioning $CLS=\{cl_{1},\ldots ,cl_{m}\}$ based on cl and $CLS_{id}$*successive* if there is a partial order of ≺ on CLS such that, if a sequential program $u^{\prime }$ is the ordered sequence $cl_{k_{1}},\ldots ,cl_{k_{m}}$ of clusters from CLS with regard to ≺, then the execution of $u^{\prime }$ is equivalent to the execution of *u*. If the final partitioning is successive, we can distribute process clusters on the corresponding number of controllers, keeping the program operational semantics.

## 5. Case-Study: Bottle-Filling System
and Sluice System


### 5.1. The Bottle-Filling System

The bottle-filling system is considered as a case study (Figure 3). The conveyor is used to transport bottles. The liquid is sterilized and maintained at 100 degrees Celsius. Two temperature sensors are attached to the tank to monitor the liquid temperature. The steam valve provides a superheated steam supply to the tank casing and heating of the tank. The conveyor also has a limit switch that is used to detect the presence of a bottle under the tank. At the bottom of the filling tank, there is a valve for pouring liquid into the bottle. The liquid level in the bottle is determined by a photo sensor. The system has two level sensors to control the degree of filling the tank with liquid. A valve at the top of the tank ensures that the tank is replenished with liquid. The algorithm of the operation is quite simple: we need to ensure the supply of bottles along the conveyor and the filling of these bottles with liquid, which, in turn, must be sterilized. It also assumes that, after the tank low sensor is triggered, the tank contains enough liquid to fill one bottle.

### 5.2. Process-Oriented Specification of Plant Simulator and Controller

The software can be described by means of two specifications, which are a plant simulator and controller.

The simulation algorithm is quite simple and plays an auxiliary role, so we provide only a textual description. The plant simulator program consists of six processes (Initialization, TankSim, TempSim, BottleFillingSim, ConveyorSim, SetBottle). The initial process Initialization launches the other processes and stops itself.

The TankSim process simulates the tank level. It calculates the TankLevel internal variable according to the state of the oTankFilling and oFillBottle signals and changes the state of the iHighLevel and iLowLevel sensors signal.

The TempSim process simulates the liquid temperature. It calculates the TankTemp internal variable according to the state of the oTankFilling and oSteam signals and changes the state of the iHighTemp and iLowTemp sensors signal.

The BottleFillingSim process simulates the liquid level in the bottle. If both the liquid level in the tank is greater than zero (TankLevel > 0.0) and the bottle is under the nozzle (iBottlePosition is ON), and the nozzle is open (oFillBottle is ON), then the liquid level in the bottle BottleLevel increases. Then, in accordance with the calculated level, it changes the state of the iBottleLevel.

The ConveyorSim process simulates the bottle movement on the conveyor. If there is a bottle on the conveyor and the conveyor is turned on (oConveyor is ON), then the coordinate of the bottle BottleCoord is incremented. If the bottle coordinate is within the specified limits, then the iBottlePosition signal is set. If the coordinate exceeds the specified threshold, then the bottle is removed from the conveyor.

The installation of a new bottle on the conveyor is carried out on the rising edge of the iSetBottle signal. The SetBottle process handles this event and puts the bottle on the conveyor.

Figure 4 shows a diagram of the processes interaction. The control algorithm is set by seven processes: processes (1) Initialization and (2) MainLoop, which provide the general control of the system; (3) process TankFilling, which controls filling the tank with liquid; processes (4) ForcedSterilization and (5) KeepSterilization, which provide a forced sterilization of the liquid in the tank and maintain the temperature of the liquid; the BottleFilling process (6), which controls the outlet valve for filling the liquid and filling the bottles to a predetermined level; and the NextBottle process (7), which controls the bottle conveyor and stops the conveyor when the bottle is under the outlet valve. The listing of the program in the poST language can be found at the link [55].

The initial process Initialization (Figure 5, left) ensures that the tank is filled with liquid and the filled tank is sterilized. At the beginning, the process launches the TankFilling process and changes its current state to WaitForFilling. In the WaitForFilling state, the process awaits the tank to fill with liquid by monitoring the inactive state of the TankFilling process. After filling the tank, the process starts the liquid sterilization process ForcedSterilization and enters the next state. In the WaitForSterilization state, the process waits for the end of the sterilization process, and then it starts the process of maintaining the temperature of the liquid and transfers control to the MainLoop process (starts the MainLoop process and stops itself).

The MainLoop process (Figure 5, right) alternately starts the NextBottle process and then the BottleFilling process. After filling the next bottle, the MainLoop process controls the level of water in the tank. If the tank is empty, the MainLoop process transfers control to the Initialization process (starts the Initialization process and stops itself).

The TankFilling process (Figure 6, left) starts filling the tank (sets the oFillTank output in state ON), waits for the liquid level to be reached (iHighLevel), and stops filling the tank. After this, it stops itself.

The ForcedSterilization process (Figure 6, center) starts heating the tank (oSteam), waits for the sterilization temperature to be reached (iHighTemp), and sterilizes the liquid for a minute. After the timeout, it stops itself.

The KeepSterilization process (Figure 6, right) maintains the temperature of the fluid within a predetermined range. When the iLowTemp signal appears, the tank heating (oSteam) is turned on, and when the sterilization temperature is reached (iHighTemp is ON), the tank heating is turned off. The peculiarity of this process is that it runs constantly and can only be started and stopped from outside.

The NextBottle process (Figure 7, left) starts the conveyor (oConveyor) and waits for the moment when the bottle is under the nozzle (iBottlePosition is ON). After the event, the process stops the conveyor and stops itself.

The BottleFilling process (Figure 7, right) opens the nozzle (oFillBottle) and waits for the moment when the bottle is full (iBottleLevel). After the event, the process closes the nozzle and stops itself.

Using the proposed Partitioning Algorithm for Bottle Filling Software:In the first step, the algorithm produces seven clusters identified by the process names: Initialization, MainLoop, TankFilling, ForcedSterilization, KeepSterilization, BottleFilling, and NextBottle clusters.In the second step, the algorithm adds the variables to the clusters: iLowLevel to the MainLoop cluster; iHighLevel and oFillTank to the TankFilling cluster; iHighTemp and oSteam to the ForcedSterilization cluster; iLowTemp to the Keep Sterilization cluster; iBottleLevel and oFillBottle to the BottleFilling cluster; iBottlePosition and oConveyor to the NextBottle cluster.In the third step, the algorithm merges the ForcedSterilization and KeepSterilization clusters to the ForcedSterilization cluster because the processes use the same variable.In the fourth step, the algorithm merges the Initialization and MainLoop clusters to the Initialization cluster because both processes use the same process (KeepSterilization).In the fifth step, the algorithm produces no changes because looped processes MainLoop and Initialization already belong to the same cluster. Thus, finally, we have five independent clusters separated by variables and processes.

### 5.3. Implementation on AVR Microcontrollers

Based on the clustering carried out, a distributed control system for the bottling line was implemented, consisting of four nodes (Figure 8). The Arduino Diecimila board based on the ATmega 168 microcontroller was used as a computing node. The MCP2515 CAN bus module was used for the internode communication.

We implemented the plant simulator on the Arduino boards according to the specification in the Reflex language (a process-oriented extension of the C language [56]). For visualization, a simple communication board was developed to provide a safe connection between the electrical signals of the plant simulator and the distributed control system. For comfortable work, the board has LEDs that show the high level of signals.

The first controller hosts two clusters with the Initialiation, MainLoop, and Tank Filling processes (physical signals iLowLevel, iHighLevel, oFillTank). The second controller hosts the cluster with ForcedSterilization and KeepSterilization processes (physical signals iLowTemp, iHighTemp, oSteam). In the third and fourth microcontrollers, the NextBottle and BottleFilling processes are located, respectively. For reading/writing physical signals, the regular functions of the Arduino platform (digitalRead()/digitalWrite()) were used. The Arduino code was implemented according to the specification in the Reflex language, which made it easier to manually translate into the C language. The implementation used a standard time-triggered pattern with an activation interval of 100 ms.

Interprocess communication over the CAN network was implemented as follows. When starting a process hosted on a third-party network node, the encoded number of the desired state (start state or inactive state) is written to the local copy of the process status word and the corresponding network message is sent. Upon completion of the running process, it sends a message about its transition to the inactive state. Upon receiving the message, the calling process sets the local copy of the current state to an inactive state. Such a protocol allows us to keep the standard implementation of inter-process communication, regardless of the location of processes in a distributed system. However, it should be noted that the protocol does not handle network failures. In real projects, this shortcoming can be eliminated either by duplicating network equipment or by introducing a special failure-handling procedure into the protocol and control algorithm.

Performance measurements were made in a standard way by measuring the execution time of 1000 function calls. The standard micros() function was used to measure the execution time. The measurements showed the following. The reading of signals by the digitalRead() function takes time in the range of 3.9–5.0 μs. Writing a signal with the digitalWrite() function takes 4.8 μs. Sending a message via CAN-bus takes 335 μs, and reading a message takes 23 μs. The execution time of the control algorithm itself does not exceed 4 μs (4 µs is the resolution of the micros() function). The worst-case execution time (WCET) for the first processor was no more than 1 ms. In general, the measurement results are typical for the industrial control systems: the most time-consuming operations are the functions of reading/writing discrete and analog signals and network messaging. The relation between WCET (1ms) and the activation interval (100 ms) ensures that there are no race conditions and thus guarantees determinism.

### 5.4. The Sluice Control System

We also apply the proposed approach to the sluice case study. A sluice (Figure 9) is a hydraulic structure on navigable waterways that ensures the passage of vessels between two water basins with different water levels. The sluice consists of a sealed lock chamber with an adjustable water level. The chamber is limited by two gates(openHighGate/openLowGate)on both sides and has dimensions sufficient to accommodate one vessel. The water level in the lock chamber is adjusted with two valves (openHighValve/openLowValve). The lock gates at the ends of the chamber serve for sealing the chamber during the passage of ships. The HighGateOpened, LowGateOpened, HighGateClosed, and LowGateClosed sensors are used to control the gate positions.

Valves are used to fill and empty the chamber. The principle of sluice operation is as follows: first, the entrance gate is opened, and the vessel enters the chamber and moors to the bollards. The entrance gate then closes and the bypass valve opens, causing the water level in the lock chamber containing the ship to fall or rise. The HighValveOpened, LowValveOpened sensors are used to monitor the state of the bypass valves. Opening the bypass valve causes the water level in the lock chamber to align with the water level in the adjacent reservoir in the direction of the vessel passage. Sensors (atLow and atHigh) are used to control the leveling of the water levels in the chamber and in the pool. Once the water reaches the required level, which is equal to the one outside the exit gate of the chamber, the exit gate opens, and a signal (Low2ChmbrLight, Chmbr2LowLight, Chmbr2HighLight, High2ChmbrLight) is given through the semaphore, allowing the vessel to continue and exit the lock. Only one ship can pass through the lock at once. The airlock is equipped with a ship presence sensor at the lower level (shipInHigh), the upper level (shipInLow), and the lock chamber (shipInChmbr).

The control algorithm was implemented using 18 processes. A detailed description of the poST-program is given in [57]. For this program, the proposed partitioning algorithm gives three clusters only. Processes in one cluster are highlighted in the same color (Figure 10). We marked with red dotted lines shared variables that formally cause the processes to be combined into one large cluster. All these variables are reading sensors only. They values are always analyzed to eliminate emergency situations at the local level. For example, the OpenLowGate process checks the atLow sensor to avoid opening the gate when the water levels are different because the acting leads to an accident.

Analyzing this confusing partitioning result, we observe the following facts. One the one hand, using these sensor variables does not necessary result in process synchronization because an asynchronized reading of the variables does not result in nondeterminism. On the other hand, when the system starts from the specified normal initial state, undesirable nondeterminism does not arise. Thus, there is no need to combine the processes into one cluster.

Following these reasoning, we removed such variables from the joining condition of our partitioning algorithm for the sluice program and obtained a more distributed system with six clusters (Figure 11).

This case study demonstrates that the outcome of partitioning is influenced by the selected architecture and implementation strategy. Additionally, we conclude that the partitioning algorithm can be improved by including a careful analysis of variable access order.

## 6. Discussion and Conclusions

The paper presents the concept of a topologically independent specification of distributed control algorithms based on the process-oriented programming paradigm and a straight approach to its implementation based on a network of microcontroller nodes. In this schema, only operations for reading input and writing output signals are physically parallelized, and the algorithm itself is executed sequentially, similarly to a centralized implementation, with synchronization of information messages and the order of execution over the bus. The approach constructively ensures the absence of data races and the preservation of the semantics of the original process-oriented program, which means the applicability of existing methods for verifying centralized process-oriented programs to distributed process-oriented programs. This opens the door to the formal verification of distributed programs using already developed approaches. In particular, our model checking method for process-oriented programs [58,59] can be applied with minor corrections that concern constructing several independent action lines of parallel processes with regard to given clusters. We also adapted our process-oriented deductive verification methods [15,60] for distributed control algorithms by applying them to each cluster separately with the original program annotations. A natural limitation on the degree of parallelization is the number of processes in the specification.

We also proposed an algorithm for splitting a process-oriented specification into clusters, which eliminates the need for data synchronization. The study of the algorithm on the “bottle-filling system” case study showed a system control algorithm that consisted of seven processes divided into five weakly connected clusters. The clusters communicate only with 11 messages related to the events of beginning and end of technological operations “sterilization”, “installation of a new bottle”, and “filling the bottle with liquid”. Clusters can be arranged in any arbitrary combination on the computing nodes of a distributed system. In any combination, the physical signals are processed locally and the implementation does not require data synchronization.

The result obtained shows the promise of using the process-oriented languages for creating distributed control software. In the final partitioning, it is possible that some processes using intersecting sets of variables cannot be active simultaneously, and therefore do not lead to non-determinism. Thus, we can relax the clustering requirements by replacing the conditions “use the same variable (or process)” with the condition “use the same variable (or process) at the same time”.

Based on the adjusted requirements, we can find potential candidates for additional clustering in the bottle-filling controller specification; for example, the activity of the ForcedSterilization and KeepSterilization processes are mutually exclusive. However, formally, this splitting cannot be performed because the MainLoop process stops the KeepSterilization process and starts the Initialization process, which, in turn, starts the ForcedSterilization process. If we assume that the frequency of the activation of processes on different nodes can be arbitrary, then non-determinism may arise: the ForcedSterilization process may already be running, but the KeepSterilization process has not yet been stopped. The resolution of this issue involves the use of additional information. Such a requirements correction obviously complicates the analysis of the specification; however, it allows for more fine-grain partitioning.

In the future, we plan to improve the partitioning algorithm by adding procedures for a more detailed analysis of access to variables. Such improvements can provide, in particular, an exception from the clustering rules for cases of sequential processing; that is, conflict-free access. Communication failure handling may involve the development of an application layer of the CAN protocol, which includes message acknowledgment and the control of response timeouts. Increasing the robustness of the system in the event of a wire break or failure of individual nodes can be achieved by introducing a watchdog mechanism and developing strategies for changing the state of the plant to a safe one. We will also conduct research on the issues of the automatic partitioning of the process-oriented algorithm for various initial conditions.

## Figures and Tables

**Figure 1 sensors-23-06216-f001:**
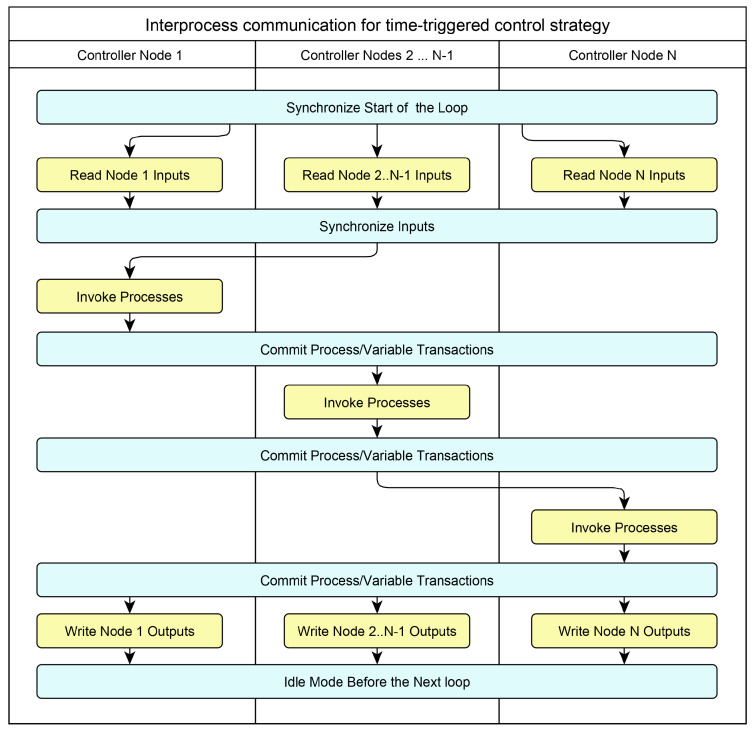
Preserving the hyperprocess semantics while deploying a process-oriented specification on a distributed architecture.

**Figure 2 sensors-23-06216-f002:**
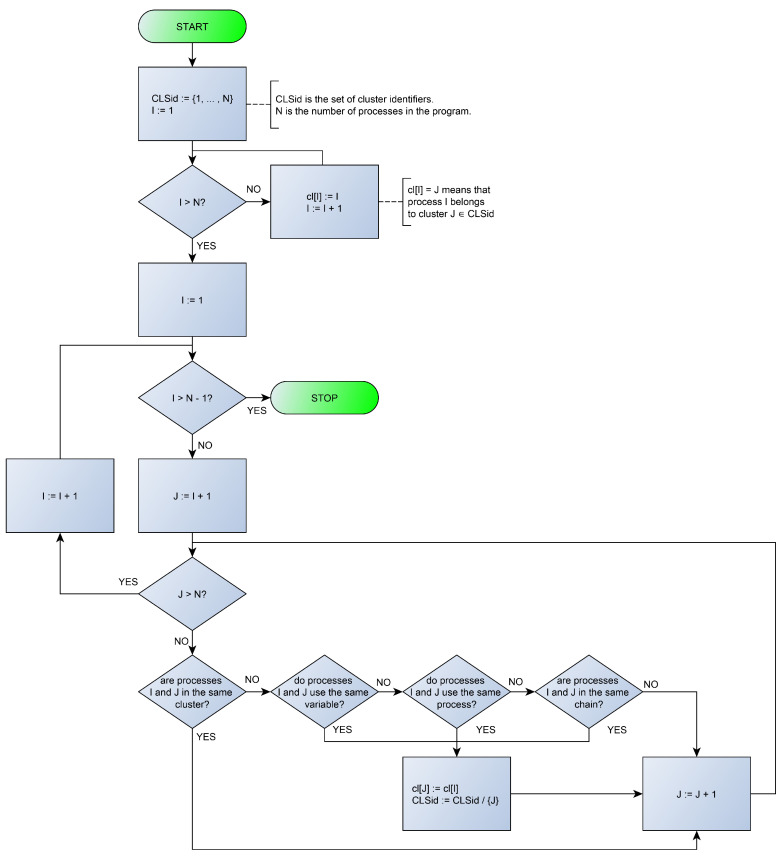
Partitioning algorithm flowchart.

**Figure 3 sensors-23-06216-f003:**
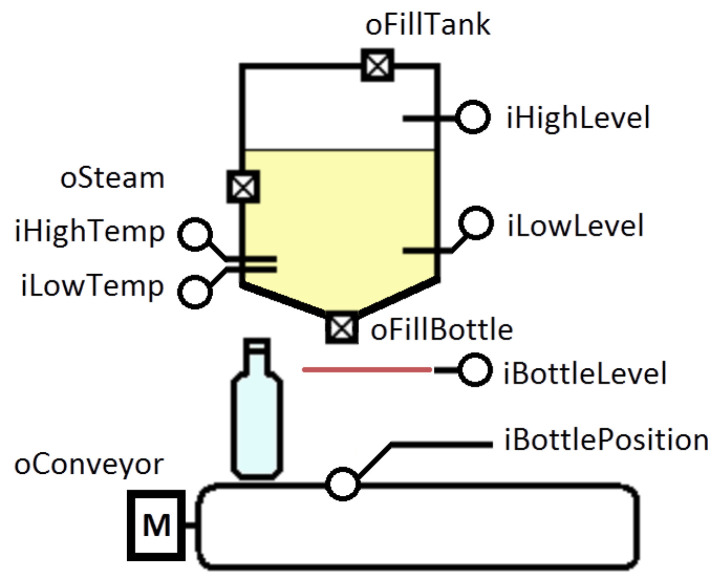
The bottle-filling system.

**Figure 4 sensors-23-06216-f004:**
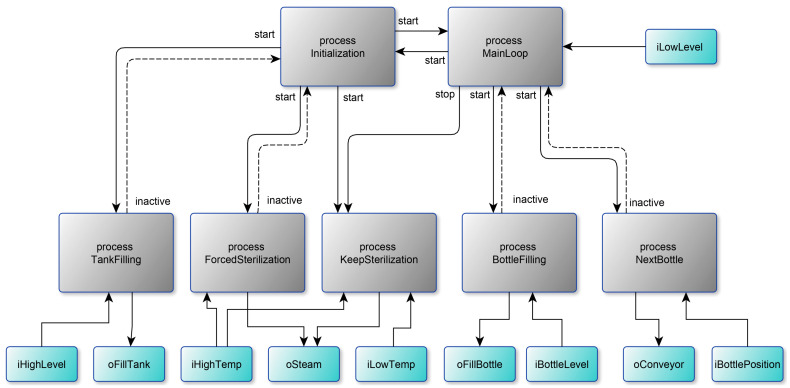
Process interaction diagram for bottle-filling software.

**Figure 5 sensors-23-06216-f005:**
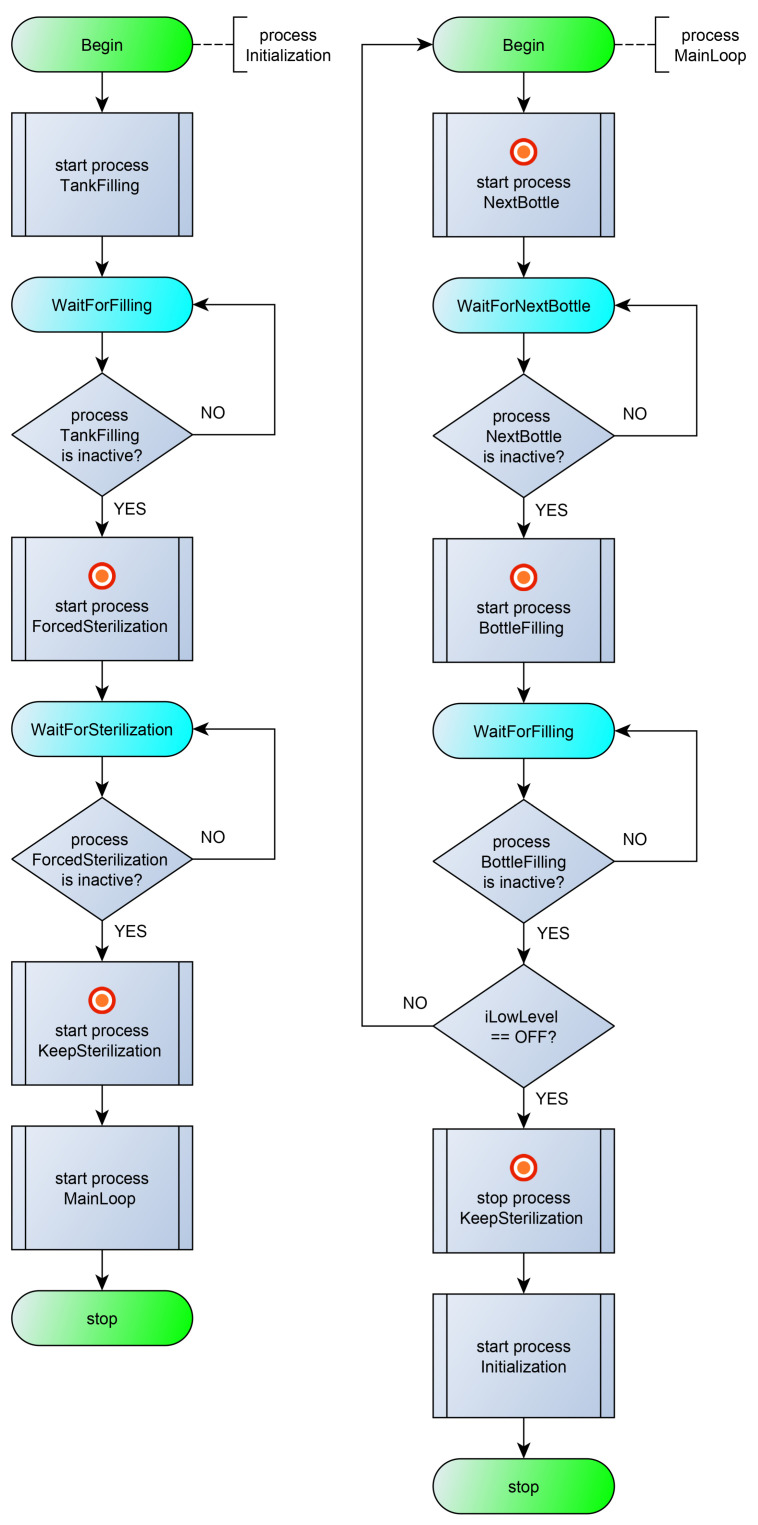
Flowcharts for the Initialization process (**on the left**) and MainLoop process (**on the right**).

**Figure 6 sensors-23-06216-f006:**
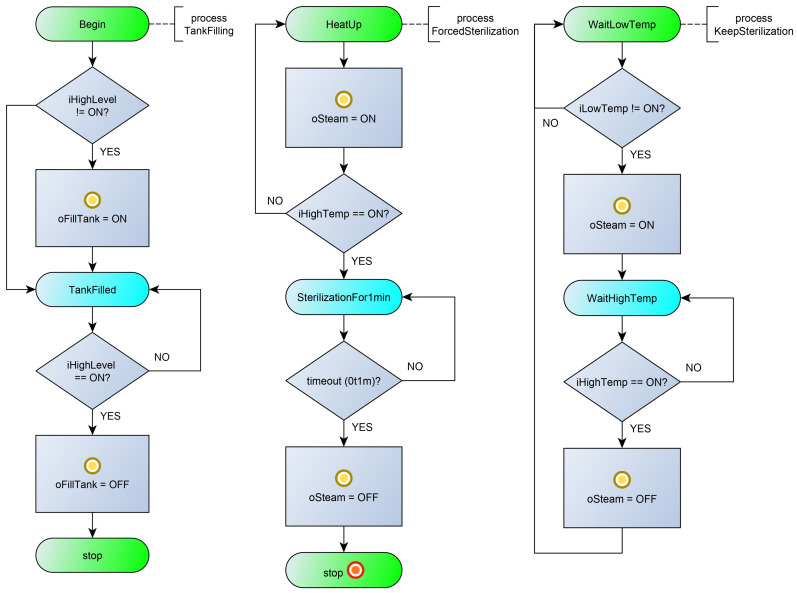
Flowcharts for the TankFilling process (**on the left**), ForcedSterilization process (**on the center**), and KeepSterilization process (**on the right**).

**Figure 7 sensors-23-06216-f007:**
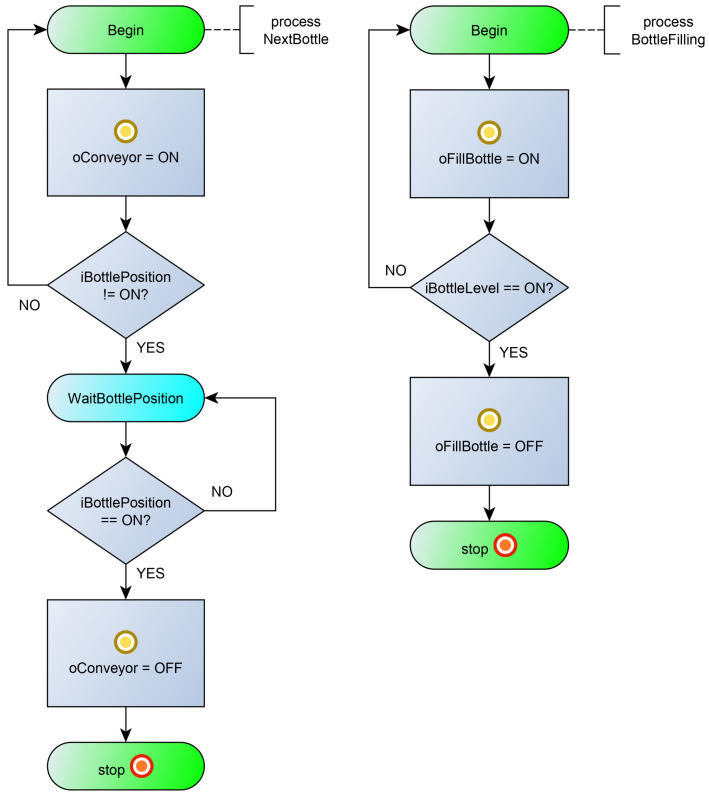
Flowcharts for the NextBottle process (**on the left**) and BottleFilling process (**on the right**).

**Figure 8 sensors-23-06216-f008:**
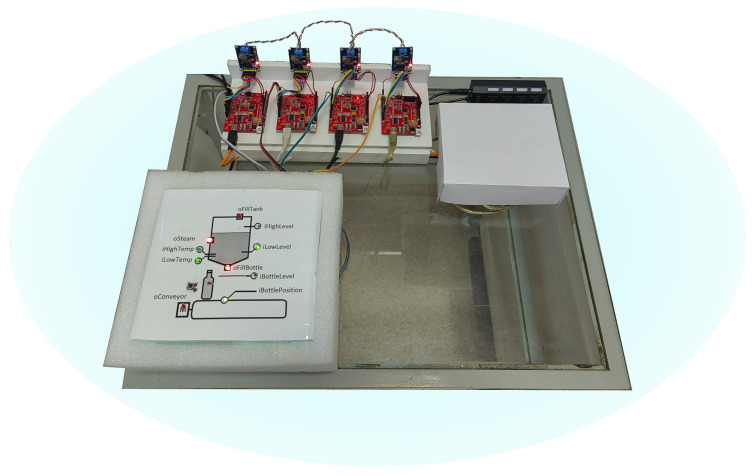
Implementation of the bottle filling system on four controllers.

**Figure 9 sensors-23-06216-f009:**
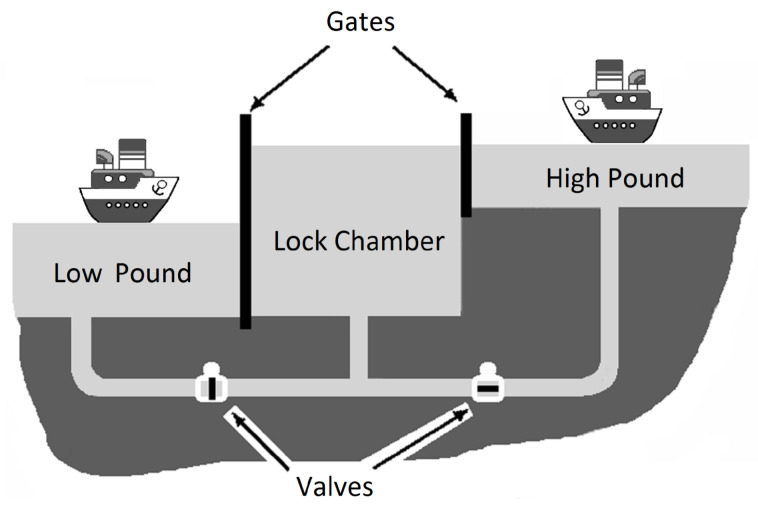
The sluice control system.

**Figure 10 sensors-23-06216-f010:**
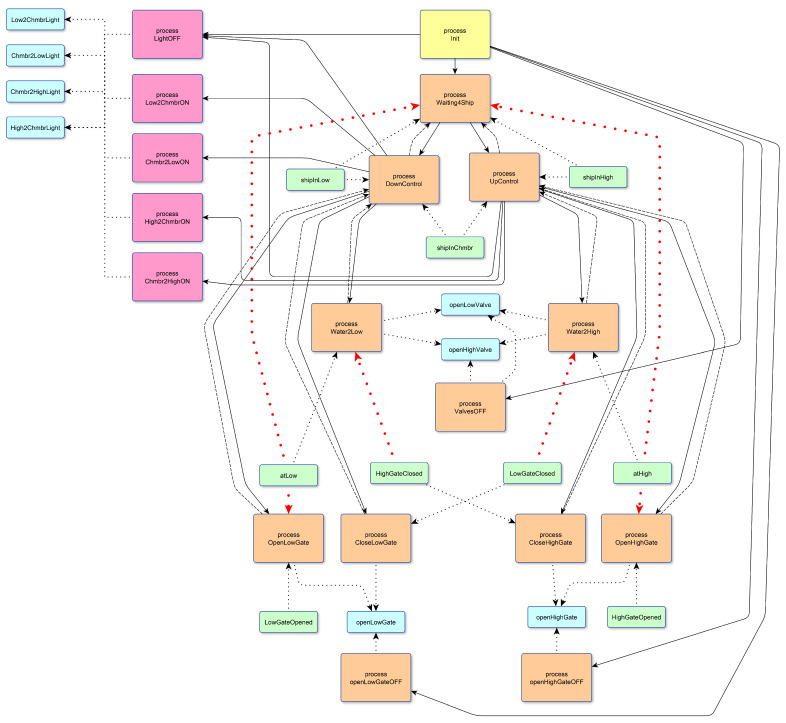
The result of the proposed partitioning algorithm for the sluice control system: the processes form three clusters, highlighted with magenta, yellow and orange.

**Figure 11 sensors-23-06216-f011:**
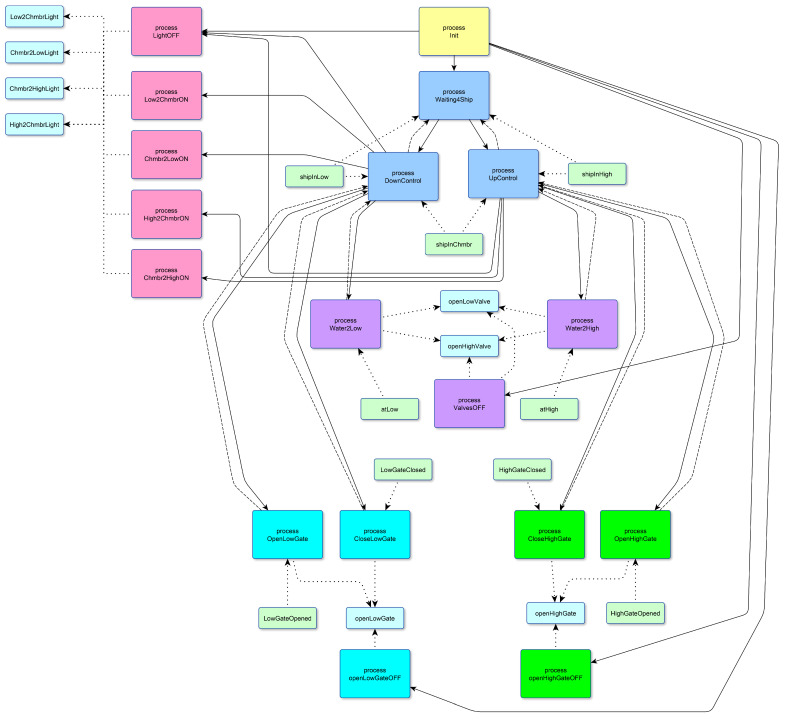
The result of the proposed partitioning algorithm ignoring safely shared variables for the sluice control system: the processes form six clusters, highlighted in magenta, yellow, blue, purple, cyan, and green.

## Data Availability

Not applicable.

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
