# Peer review of "Towards Topology-Free Programming for Cyber-Physical Systems with Process-Oriented Paradigm"

_sensors, 2023, doi:10.3390/s23136216_

Round 1

Reviewer 1 Report

The paper significantly contributes to distributed control systems and introduces an innovative concept of topology-free specification using a process-oriented programming paradigm. The proposed approach highlights the potential advantages of using process-oriented languages for creating distributed control software.

The paper is an innovative piece of research that builds upon the existing knowledge of process-oriented programming to provide a new perspective on distributed control systems. The primary strength of the paper lies in its approach to delivering a topology-free specification of control algorithms. This concept could revolutionize the way distributed control systems are designed and operated. Additionally, presenting a standard heuristic algorithm to partition a sequential process-oriented program into independent clusters is a notable contribution.

Including the "bottling system" case study strengthens the paper by offering a practical algorithm illustration. This demonstrates the feasibility of the concept and makes the paper's technical content more accessible to a broad range of readers.

However, a few areas could use further refinement:

1.     Greater Detail on Algorithm: The paper presents a heuristic algorithm for partitioning a sequential process-oriented program into independent clusters. While the paper outlines the algorithm's benefits, there is a lack of a detailed explanation or walk-through of the algorithm. The paper could be improved by providing a step-by-step guide or pseudo-code to help readers understand how this algorithm works in practice. A more detailed presentation of the clustering algorithm, including step-by-step explanations or flow diagrams, could enhance reader comprehension.

2.     Broaden the Scope of Case Studies: Although insightful,usingf a single case study limits the ability to extrapolate the algorithm's effectiveness across various scenarios. Additional case studies across diverse applications would strengthen the argument for the algorithm's generalizability and robustness. Using a single case study, the “Bottling system,” is an excellent start. Still, adding more case studies across different sectors would strengthen the paper's assertions and provide a more comprehensive understanding of the proposed algorithm's applicability.

3.     Detailed Analysis of Limitations: The authors acknowledge the issue of non-determinism when processes on different nodes can be activated at an arbitrary frequency. However, the paper would benefit from a more comprehensive exploration of such limitations and challenges, possibly even dedicating a section to this. The paper mentions potential issues like non-determinism when processes using intersecting sets of variables cannot be active simultaneously. While the paper proposes a solution (relaxing the clustering requirements), a deeper exploration of this issue, with examples demonstrating how this might affect real-world scenarios, would strengthen the paper.

4.     In-depth Discussion on Verification Methods: The paper asserts that existing formal verification methods can be used, given that the semantics of the original process-oriented program are preserved. However, an in-depth discussion or a demonstration of how these methods can be applied would be beneficial. Give concrete examples or simulations demonstrating how formal verification methods can be used in this context.

5.     More Detailed Explanation of Clustering Algorithm: While the paper proposes an algorithm for splitting a process-oriented specification into clusters, it might benefit from a more comprehensive explanation or a graphical depiction of the algorithm. This would help readers better understand the process.

6.     The authors mention they plan to research handling failures and wire breaks in the future, but an initial discussion on these topics would have been valuable to include, even if only to discuss potential strategies or theories. Applying the proposed algorithm to other practical and diverse scenarios would provide additional proof of concept and showcase its versatility. Provide a more extensive discussion on potential hurdles in implementing the proposed system, such as synchronization issues and possible solutions or workarounds. Expand the future work section to discuss more specific strategies for handling automatic partitioning, failures, and wire breaks. This will give the reader a clear picture of this field's potential growth and development.

Author Response

Answers of the authors to the reviewer comments.

We are very grateful to the anonymous reviewer who provided us with very insightful and helpful comments and recommendations. In the revised version of the paper, we have addressed all comments and recommendations of the reviewers, as described in the following. We list the comments of the reviewer (marked with R) and our answers marked with A. We highlighted all our additions to the manuscript in green.

R1. Greater Detail on Algorithm: The paper presents a heuristic algorithm for partitioning a sequential process-oriented program into independent clusters. While the paper outlines the algorithm's benefits, there is a lack of a detailed explanation or walk-through of the algorithm….

A1. We have provided the pseudocode of our partitioning algorithm, describe in detail its steps and give algorithm flowchart (lines 440-448 of the paper). 

R2.     Broaden the Scope of Case Studies: Although insightful, using a single case study limits the ability to extrapolate the algorithm's effectiveness across various scenarios. Additional case studies across diverse applications would strengthen the argument for the algorithm's generalizability and robustness….

A2. We have added a new Subsection 5.4 which describes application of our partitioning  method to a Sluice System case study.  (lines 589-627 of the paper). The result of the clustering of the sluice control algorithm shows a clear direction of improving our partitioning algorithm to achieve better distributivity.

R3.     Detailed Analysis of Limitations: The authors acknowledge the issue of non-determinism when processes on different nodes can be activated at an arbitrary frequency…

A3. We discuss the limitation and improving direction of our current approach in the new Subsection 5.4  (lines 618-627 of the paper). We have pointed out that a specific structure of using variables and processes in a control algorithm affects a partition quality in a sense of more or less distributivity.

R4.     In-depth Discussion on Verification Methods: The paper asserts that existing formal verification methods can be used, given that the semantics of the original process-oriented program are preserved. However, an in-depth discussion or a demonstration of how these methods can be applied would be beneficial. Give concrete examples or simulations demonstrating how formal verification methods can be used in this context.

A4. We have described the ideas of distributive adaptation of our formal verification methods developed earlier for non-distributed process-oriented programs in lines 638-642 of the paper.

R5.     More Detailed Explanation of Clustering Algorithm: While the paper proposes an algorithm for splitting a process-oriented specification into clusters, it might benefit from a more comprehensive explanation or a graphical depiction of the algorithm.

A5. We have provided the flowchart of our partitioning algorithm in Fig. 2, page 10. 

R6. Failures and wire breaks:     The authors mention they plan to research handling failures and wire breaks in the future, but an initial discussion on these topics would have been valuable to include, even if only to discuss potential strategies or theories.

A6. We have proposed some intuitions about our vision of handling wire breaks and other failures of controller nodes to provide robustness of distributed control algorithms in lines 667-673 of the paper.

We hope that these improvements will help understanding and readability of our paper. Thanks a lot for your comments and suggestions.

Reviewer 2 Report

The paper tackles topology-free development of programs solving the distributed control system problems.

Some of these main problems are presented and their used solutions are discussed. The applications where this approach is used are presented.   

The main contribution consists of the conceiving of an algorithm for partitioning the application in clusters that fulfill some properties.  

A practical application of bottle-filling system is given for method experimentation.  The article is well organized and written, the language is clear, the included figures sustain the understanding of the proposed method.

A good review of the distributed control systems is included. Their main properties are analyzed.  

The method description is clear and so is the algorithm.

Adding a formal verification for fulfilling of the distributed systems’ properties would increase the proposal quality.

The experiment is well chosen, and it sustains the conclusions.

 Please verify the Line: 198.

Author Response

Answers of the authors to the reviewer comments.

We are very grateful to the anonymous reviewer who provided us with very insightful and helpful comments and recommendations. In the revised version of the paper, we have addressed all comments and recommendations of the reviewers, as described in the following. We list the comments of the reviewer (marked with R) and our answers marked with A. We highlighted all our additions to the manuscript in green.

R. Please verify the Line: 198.

A. Thank you very much, we have corrected the text.
